# High-Speed and Low-Load Twin-Roll Casting of Al–5%Mg Strip

**Kazuki Yamazaki** [1] **and Toshio Haga** [2],*

1  Graduate School, Osaka Institute of Technology, 5-16-1 Omiya Asahiku, Osaka 535-8585, Japan
2  Department of Mechanical Engineering, Osaka Institute of Technology, 5-16-1 Omiya Asahiku, Osaka 535-8585, Japan
*  Correspondence: toshio.haga@oit.ac.jp

**Abstract:** A conventional twin-roll caster usually casts strips at a speed slower than 2 m/min and a roll load greater than 2 kN/mm. A vertical-type high-speed twin-roll caster can cast aluminum alloy strips at a speed higher than 10 m/min and a roll load smaller than 550 N/mm due to the effect of the large thermal conductivity of copper alloy rolls, compared with that of steel rolls. However, the properties of an aluminum alloy strip cast at a roll load smaller than 140 N/mm and at a roll speed higher than 30 m/min are not clear. In this study, Al–5%Mg strips were cast at a roll speed of 30 m/min and roll loads of 2 and 88 N/mm using a vertical-type high-speed twin-roll caster. The effects of roll load on cracking at the strip surface, the tensile mechanical properties, and the microstructure were investigated. Rotating rolls were stopped during casting, and the progression of centerline segregation and the microstructure between the rolls was investigated. Centerline segregation and surface cracking were lower at 2 N/mm than at 88 N/mm. In the strip cast at 2 N/mm, elongation in the width direction was greater than that cast at 88 N/mm, due to decreased surface cracking. The results demonstrate that a very low roll load improves the properties of the cast strip.

**Keywords:** vertical-type high-speed twin-roll caster; Al–Mg alloy; roll load; center segregation; surface crack; elongation

## 1. Introduction

A conventional twin-roll caster for aluminum alloys (CTRCA) has the advantages of rapid solidification and a short processing time [1–6]. However, this caster also has disadvantages [1–6], such as low productivity and the output being marred by centerline segregation and ripple marks. The roll load of CTRCAs was increased in the 1990s to increase the roll speed. However, the roll speed is still usually lower than 2 m/min, and the roll load is greater than 2 kN/mm. The roll speed can be increased by using a longer cooling zone (setback length) and increasing the thickness reduction at the roll bite, causing the strip to become thinner. This allows the strip to be sufficiently cooled at higher roll speeds. The larger roll load also increases heat transfer between the roll and the solidified layer, which is useful for cooling the strip. CTRCA rolls are made from steel.

A vertical high-speed twin-roll caster (VHSTRC) has also been proposed to further increase the roll speed [7,8]. The casting speed of a VHSTRC ranges from 10 to 90 m/min. The method for increasing the roll speed of a VHSTRC is different from increasing that of a CTRCA. The roll material in a VHSTRC is copper or copper alloy, which increases the cooling ability due to higher heat transfer between the roll and the molten metal/solidified layer. The thermal conductivity of copper is about eight times higher than that of steel, and the increase in temperature for a copper roll is lower than that of a steel roll when in contact with the molten metal/the solidified layer. This leads to increased heat transfer between the copper roll and the solidified layer. In a CTRCA, a parting material is sprayed

on the steel roll to prevent the strip from adhering to the roll. A parting material is not required for a copper roll because of its lower surface temperature. The parting material creates thermal resistance between the roll and the molten metal/solidified layer. The roll load for a CTRCA is usually larger than 2 kN/mm, whereas that of a VHSTRC is usually from 140 to 550 N/mm. Thus, by using copper or a copper alloy, high-speed roll casting is possible at a smaller roll load [9]. In a VHSTRC, heat transfer between the roll and molten metal/solidified layer does not increase with increasing roll load.

In strips cast with a twin-roll caster, cracking, ripple marks on the surface, and centerline segregation typically occur. There have been a number of experimental and numerical studies on surface crack formation on strips formed using twin-roll casters [10–13]. However, there have been few experimental studies on surface cracking for aluminum alloy strips cast using a VHSTRC. In our previous study, we found that rolls with grooves were useful for eliminating surface cracks on 6000-series Al–Mg–Si strips [14]. However, such rolls were ineffective for suppressing surface cracks on Al–Mg strips cast using a high-speed twin-roll caster. The presence of grooves improves the uniformity of the contact between the roll and the molten metal. When this contact is poor, a delay in solidification occurs, and the strip thickness decreases. Cracks occur in thinner regions of the strip due to tensile stress at the roll bite. For Al–Mg alloys, the improvement of a delay in solidification is not sufficient for suppressing surface crack formation [15]. Therefore, this study focused on the effect of the roll load. For Al–Mg strips, surface cracks do not occur when the strip is cast using a single-roll caster equipped with a scraper [16]. The pushing load applied to the solidified layer due to the scraper is very small, and this is thought to suppress surface cracks. These defects are common in Al–Mg alloys, especially when the Mg content is greater than 3%. A previous study by our team on a VHSTRC made it clear that surface cracking, ripple marks, and centerline segregation easily occur in Al–Mg alloys [17]. In this study, strips of Al–5%Mg were cast with a smaller roll load than is usual for a VHSTRC, and the effect on defect formation was investigated. The 5083 and 5182 Al–Mg alloys are popular for sheet forming, and their Mg content is about 5%. For these reasons, Al–5%Mg alloy was used in this study. In a VHSTRC, casting has not been conducted at roll loads smaller than 140 N/mm, and the properties of the resulting strips are not clear. Specifically, the effect of a very small roll load on cracking, ripple mark formation, and centerline segregation is unknown. In this study, roll casting was performed at roll loads of 2 and 88 N/mm, and the effect of the roll load was investigated.

The cross section of a strip cast using the VHSTRC consists of three layers [18,19]. The center layer is referred to as a "band area", and it is typical for strips cast with a VHSTRC. The effect of a small roll load on the band area was investigated in this study. Changes in the microstructure of the band area near the roll bite were investigated by the sudden stopping of roll rotation (roll-stop). Weck's reagent was used to evaluate the solution state of the Mg after roll-stop [20].

## 2. Experimental Conditions

A schematic illustration of the VHSTRC is shown in Figure 1. The diameter and width of the copper rolls were 300 and 50 mm. No parting material was used on the roll's surface. The roll speed was 30 m/min, the solidification length was 100 mm, and the melt head was 100 mm. The initial roll load was 2 or 88 N/mm. When the roll load was smaller than 2 N/mm, the strip did not sufficiently solidify, and it could not be continuously cast. The smallest practical roll load was therefore 2 N/mm, under these conditions. The initial roll gap was 1 mm. The roll load was adjusted by changing the roll-spring compression. The rolls were rotated at the designated speed, and molten Al–5%Mg was poured from a crucible at 655 °C through a launder to cast the alloy into strips. The chemical composition of the Al–5%Mg alloy is shown in Table 1.

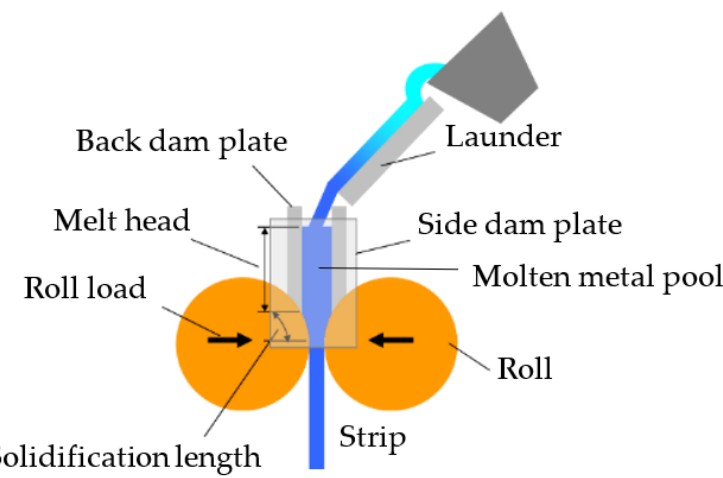

**Figure 1.** Schematic illustration of a vertical-type high-speed twin-roll caster.

**Table 1.** Chemical composition of Al–Mg alloy (mass %).

| Cu | Si | Mg | Zn | Fe | Mn | Ni | Ti | Sn | Cr | Al |
|---|---|---|---|---|---|---|---|---|---|---|
| 0.02 | 0.10 | 4.86 | 0.01 | 0.16 | 0.44 | 0.00 | 0.02 | 0.00 | 0.00 | Bal. |

During casting, the rotating rolls were stopped (roll-stop) to investigate the evolution of centerline segregation. When the roll was stopped, a chiller was inserted into the melt pool to prevent the re-melting of the solidified specimen between the rolls. The as-cast strips were cold-rolled down to plates 1 mm thick, and test pieces for tensile testing were made from these plates. The shape of the tensile test pieces is shown in Figure 2. Test pieces were cut in both the casting and width directions and annealed at 370 °C for 90 min. Penetrant testing was conducted on as-cast strips and cold-rolled 1 mm thick plates, and their surfaces were visually checked for cracks. Specimens produced by roll-stop were etched using a 5% hydrofluoric acid solution or Weck's reagent [20]. The latter was used because it is sensitive to the presence of Mg. The chemical composition was 4 g of $KMnO_4$, 1 g of NaOH, and 100 mL of distilled water. Specimen cross sections were etched and investigated using optical microscopy. A line analysis was conducted using field-emission electron-probe microanalysis (FE-EPMA, JEOL JXA-8530F, Akishima, Tokyo, Japan).

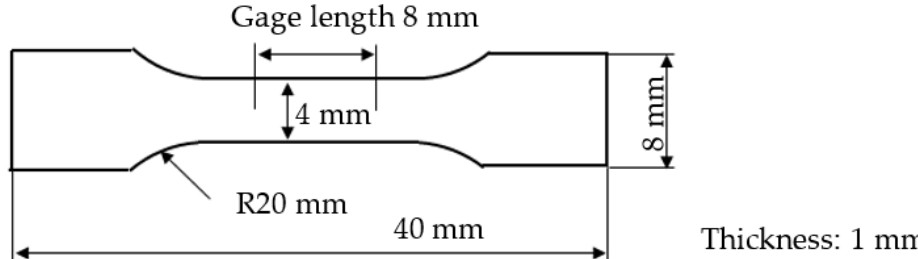

**Figure 2.** Test piece for tensile test. As-cast strip was cold-rolled down to a 1 mm thick plate and annealed at 370 °C for 90 min.

## 3. Results

### 3.1. Strip Surface

The thicknesses of strips cast at 2 and 88 M/mm were 4.8 and 4.1 mm, respectively. The surfaces of as-cast and cold-rolled strips are shown in Figures 3 and 4, respectively. The corresponding surfaces after penetrant testing are also shown, and cracks revealed by testing are shown in red. In Figure 3, surface ripple marks are seen on the as-cast strips. These marks are less evident on the strip cast at 2 N/mm than on the strip cast at 88 N/mm.

No cracks are seen on the as-cast strip cast at 2 N/mm, but some do appear on the as-cast trip cast at 88 N/mm. Thus, a small roll load of 2 N/mm decreased ripple marks and surface cracking.

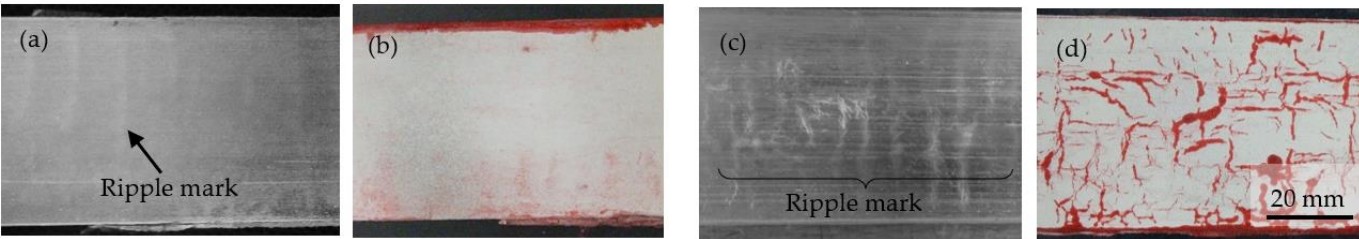

**Figure 3.** Surfaces of as-cast strips. (**a,b**): Roll load of 2 N/mm; (**c,d**): roll load of 88 N/mm; (**b,d**): penetrant tests. In the penetrant test, red indicates cracks.

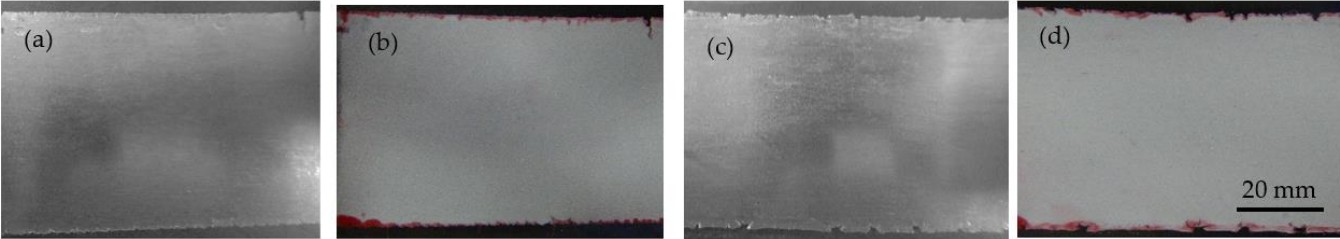

**Figure 4.** Surfaces of strips after cold rolling down to 1 mm thick plates. (**a,b**): Roll load of 2 N/mm; (**c,d**): roll load of 88 N/mm; (**b,d**): penetrant test. In the penetrant test, red indicates cracks.

No ripple marks are seen on the surface of plates cold-rolled from strips cast at 2 or 88 N/mm, as shown in Figure 4. Cracks on the plate surfaces after cold rolling could not be detected by penetrant testing. These results demonstrate that cold rolling is effective for improving the surface's finish.

*3.2. Tensile Test*

The results of tensile tests are shown in Figure 5. When the roll load is 2 N/mm, the tensile strength in the width direction is 13 MPa greater than that in the casting direction. The elongation in the width direction is about 3% smaller than that in the casting direction. When the test direction is the same as the casting direction, the tensile strength and elongation for the strip cast at 88 N/mm is almost the same as that cast at 2 N/mm. When the roll load is 88 N/mm, the tensile strength and the elongation in the width direction are much smaller than those in the casting direction.

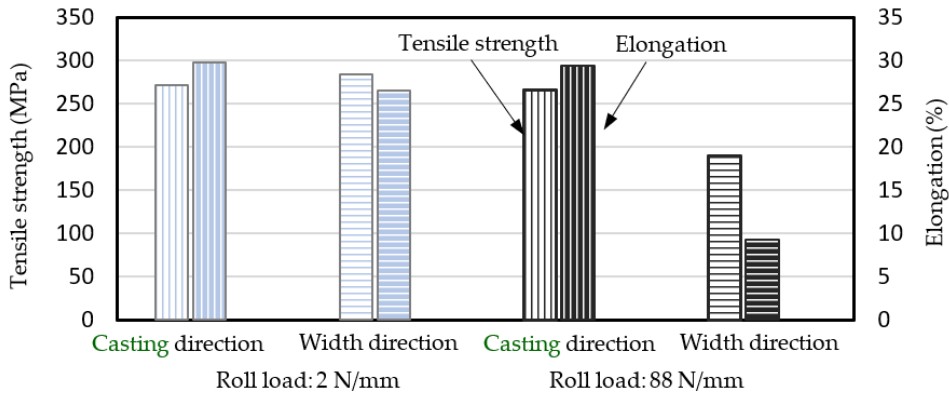

**Figure 5.** Effect of roll load and tensile direction on results of tensile tests.

### 3.3. Microstructure near Roll Bite Obtained by Roll-Stop

Cross sections near the roll bite obtained by roll-stop, following etching using a 5% hydrofluoric acid solution, are shown in Figures 6 and 7. The roll load in Figure 6 is 2 N/m, and that in Figure 7 is 88 N/mm. Both overall and enlarged views are shown in Figures 6 and 7. The band region for the strip cast at 88 N/mm is clearer than that of the strip cast at 2 N/mm.

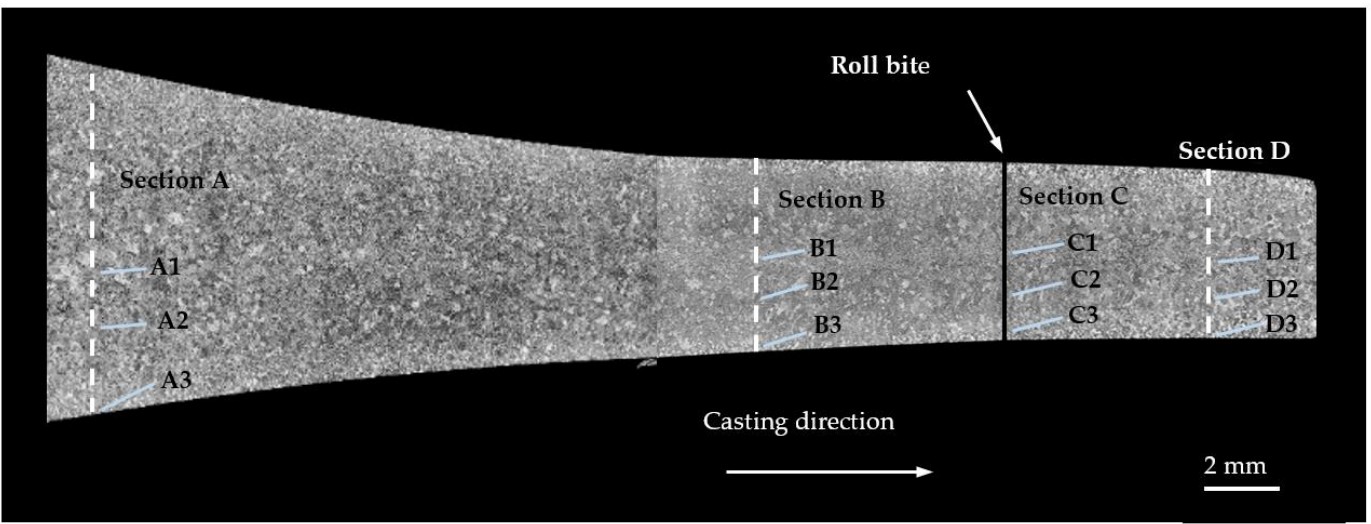

(**a**) Over all cross section

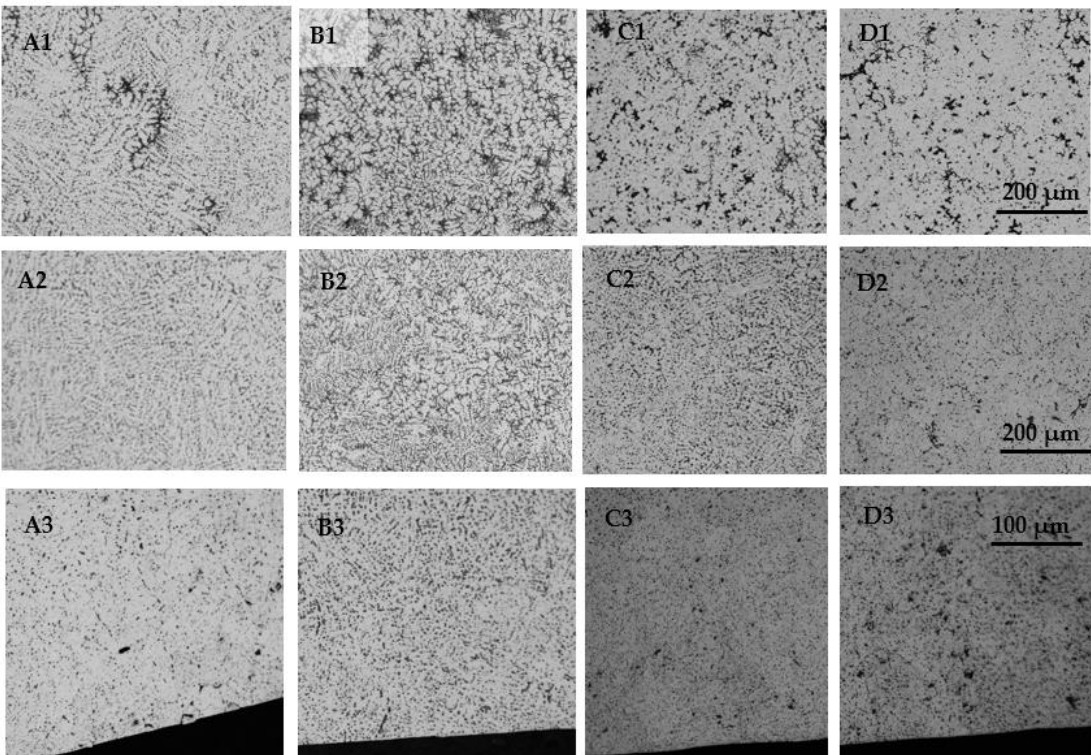

(**b**) Enlarged views of each section of (**a**)

**Figure 6.** Microstructure near roll bite obtained by roll stop. Etching: 5% of hydrofluoric acid solution. Roll load: 2 N/mm. A1–D3 in (**b**) correspond to A1–D3 in (**a**).

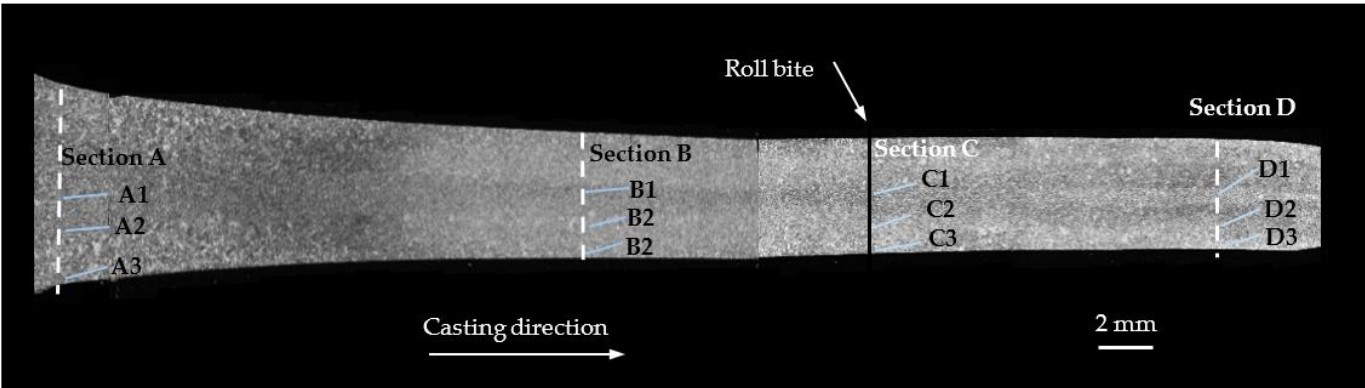

(**a**) Overall view of cross section

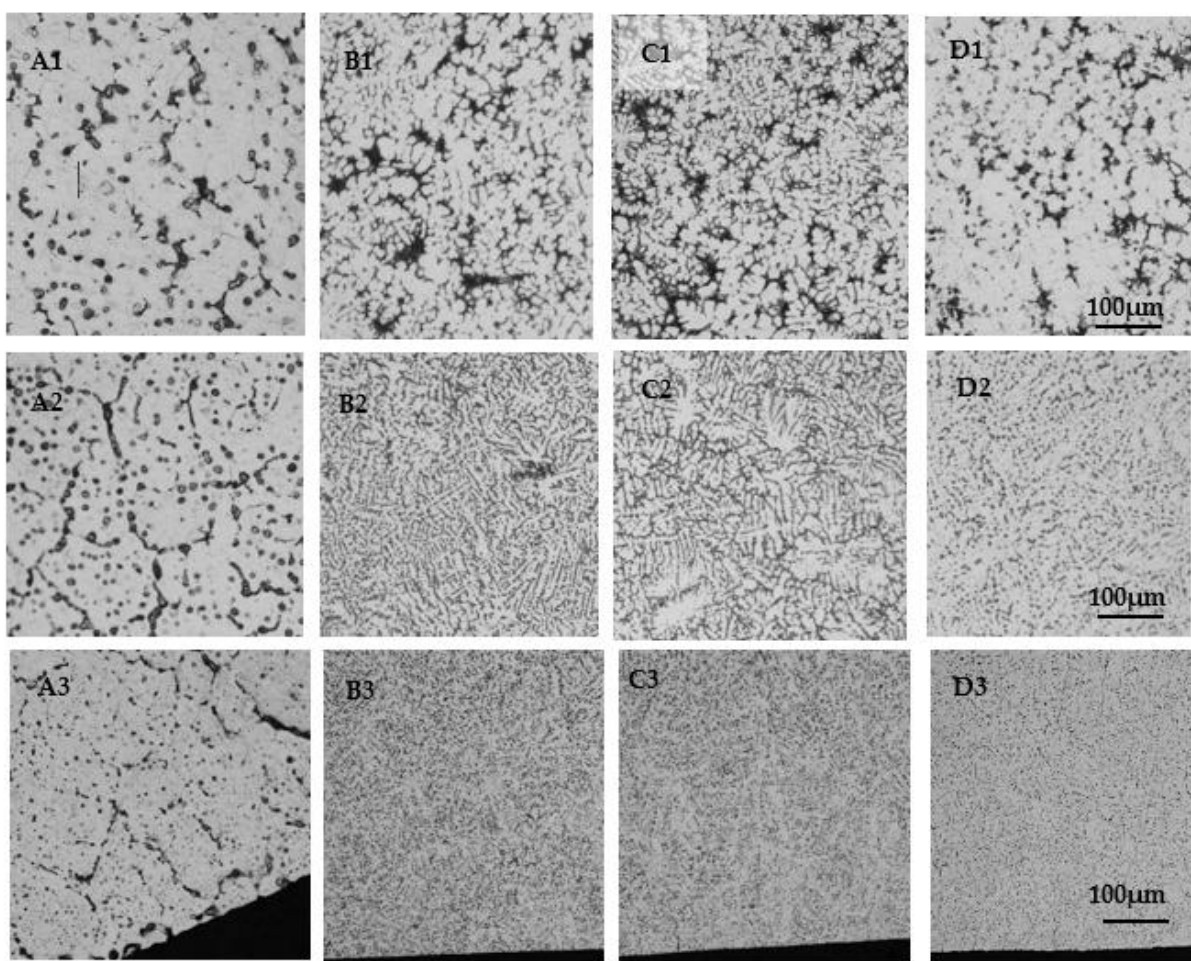

(**b**) Enlarged view of each cross section of (**a**)

**Figure 7.** Microstructure near roll bite obtained by roll stop. Etching: 5% of hydrofluoric acid solution. Roll load: 88 N/mm. A1–D3 in (**b**) correspond to A1–D3 in (**a**).

The dendrite size in Figure 6 decreases before section B but then increases. The dendrite size in Figure 7 decreases before section C (the roll bite) but then increases. Dendrites on the surface of the strip are smaller than those inside the strip (Figures 6 and 7). The dendrites change after exiting the roll bite. The dendrites in the band area after exiting the roll bite appear coarse. The dendrites on both sides of the band area look like annealed dendrites.

Cross sections obtained from roll-stop specimens following etching by Weck's reagent are shown in Figures 8 and 9 for roll loads of 2 and 88 N/mm, respectively. Regions rich in

the alloying element (Mg) appear bright, and those rich in Al appear dark [20]. The band area and surrounding solidified layers in Figure 8a are thicker than those in Figure 9a. In Figure 8b, near the surface, as shown in the enlarged views indicated by A3, B3, C3, and D3, the color is different between sections, which means that the Mg content in grains is different. In section B3, the grains are dark and the grain boundaries are bright, making a clear distinction between grains and their boundaries. In section C3, there is a higher density of precipitates at grain boundaries. In section D3, the grains are bright and their boundaries are unclear. In the middle of the thickness direction, the origin of the band area is between sections A1 and B1. The band area becomes narrower toward section C1 (roll bite). The dendrites start becoming dark in section A1 and are darkest at section B1. Bright areas with Mg enrichment appear at section B1, as shown by arrows. The Mg-rich areas in sections C1 and D1 are smaller than those in section B1. Between the strip surface and the band area, Mg-rich areas increase from sections A2 toward D2.

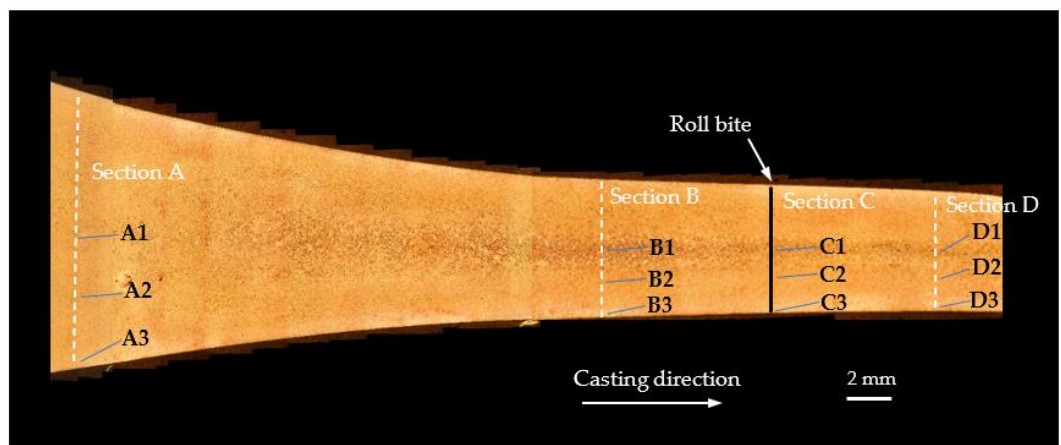

(a) Overall view of cross section

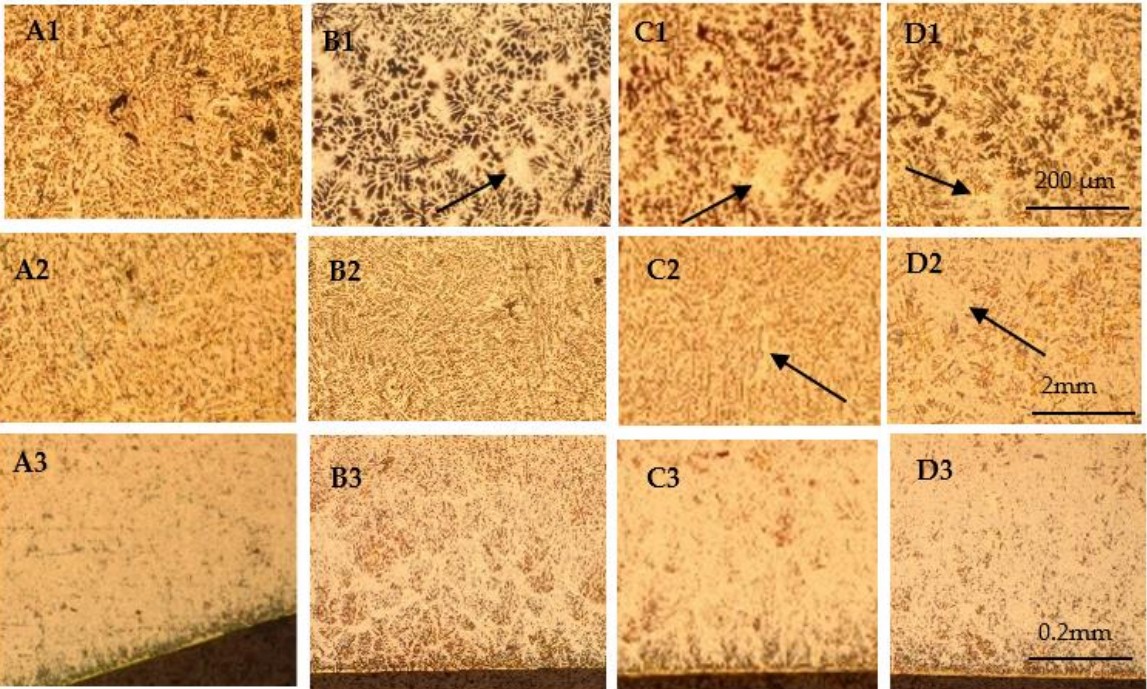

(b) Enlarged view of each cross section of (a)

**Figure 8.** Microstructure near roll bite obtained by roll stop. Etching: Weck's regent. Roll load: 2 N/mm. A1–D3 in (b) correspond to A1–D3 in (a). Arrows of (b) show Mg-rich areas.

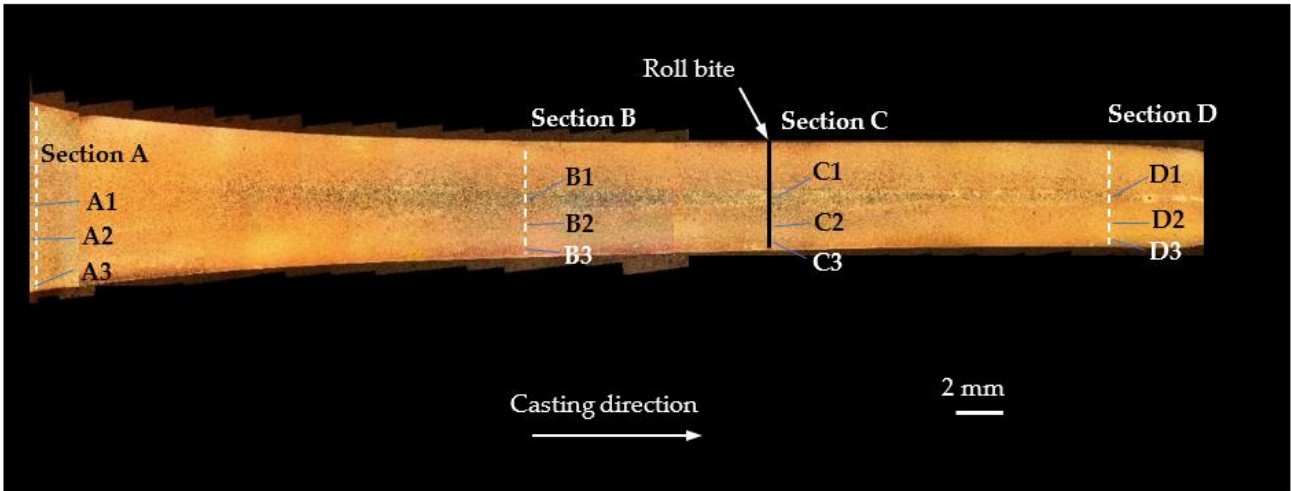

(a) Overall view of cross section

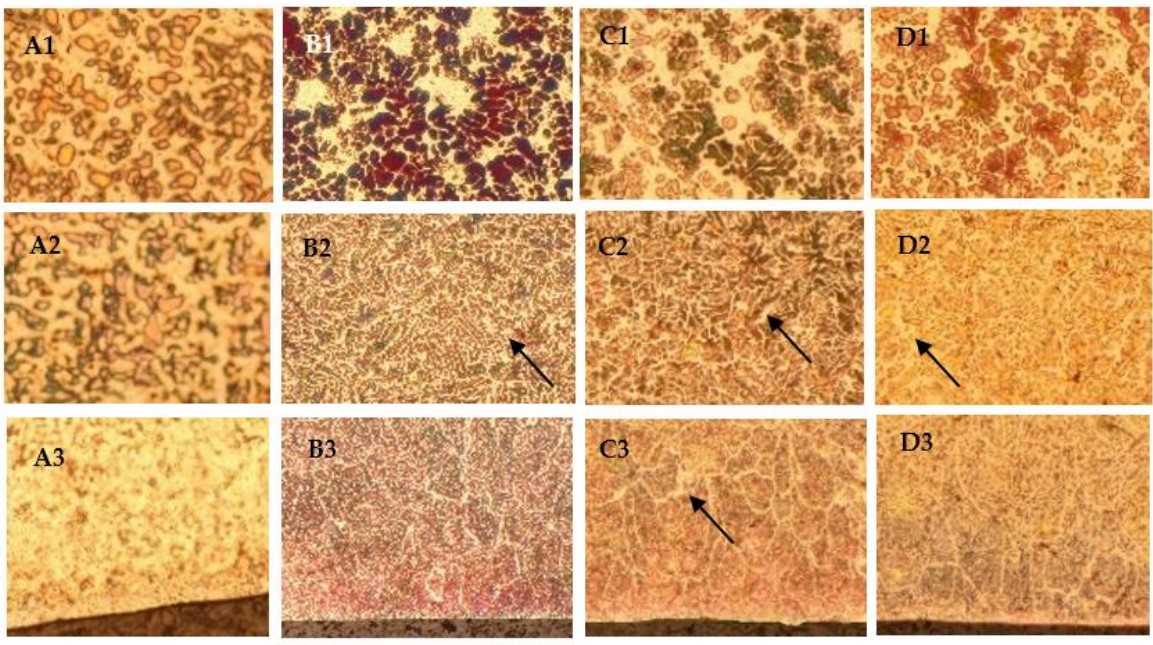

(b) Enlarged view of each cross section of (a)

**Figure 9.** Microstructure near roll bite obtained by roll stop. Etching: Weck's regent. Roll load: 88 N/mm. A1–D3 in (**b**) correspond to A1–D3 in (**a**). Arrows of (**b**) show Mg-rich areas.

The band area can be clearly seen in Figure 9a. The thickness of the solidified layers on both sides is almost uniform from section B to section D, after which it gradually decreases. The band area darkens between sections A and B, and becomes darkest between sections B and C. The dendrites become dark, as shown in B1 in Figure 9b, and as a result, the band area looks dark, as shown in Figure 9a. In section B, the Mg-rich areas are brighter than those in sections A, C, and D. In Figure 9b, no dendrites are found in sections C1 and D1 as their outlines are covered by the yellow Mg-rich area. Dendrites are found in sections C2 and D2, and grain boundaries are observed in sections C3 and D3. The grain boundaries in sections B2 and C2 are clearer than those in sections A2 and D2. Grain boundaries in sections B3, C3, and D3 in Figure 9b are clearer than those in Figure 8b.

### 3.4. Line Analysis of Mg

Figures 10 and 11 show the results of a line analysis for Mg, where sections A, B, C, and D correspond to the sections in Figures 8 and 9, respectively. When the roll load is 2 N/mm,

the Mg content decreases inside the band area and increases at its edges in sections B, C, and D, as shown in Figure 10. The Mg distribution in the band area is concave. When the roll load is 88 N/mm, the Mg content increases in the band area, and the distribution is convex in sections B, C, and D, as shown in Figure 11. The Mg distribution appears to correspond to conventional centerline segregation. The maximum Mg content in section C in Figures 10 and 11 is 6.4 and 11.0 mass %, respectively. This shows that the Mg content in the band area decreases when the roll load decreases.

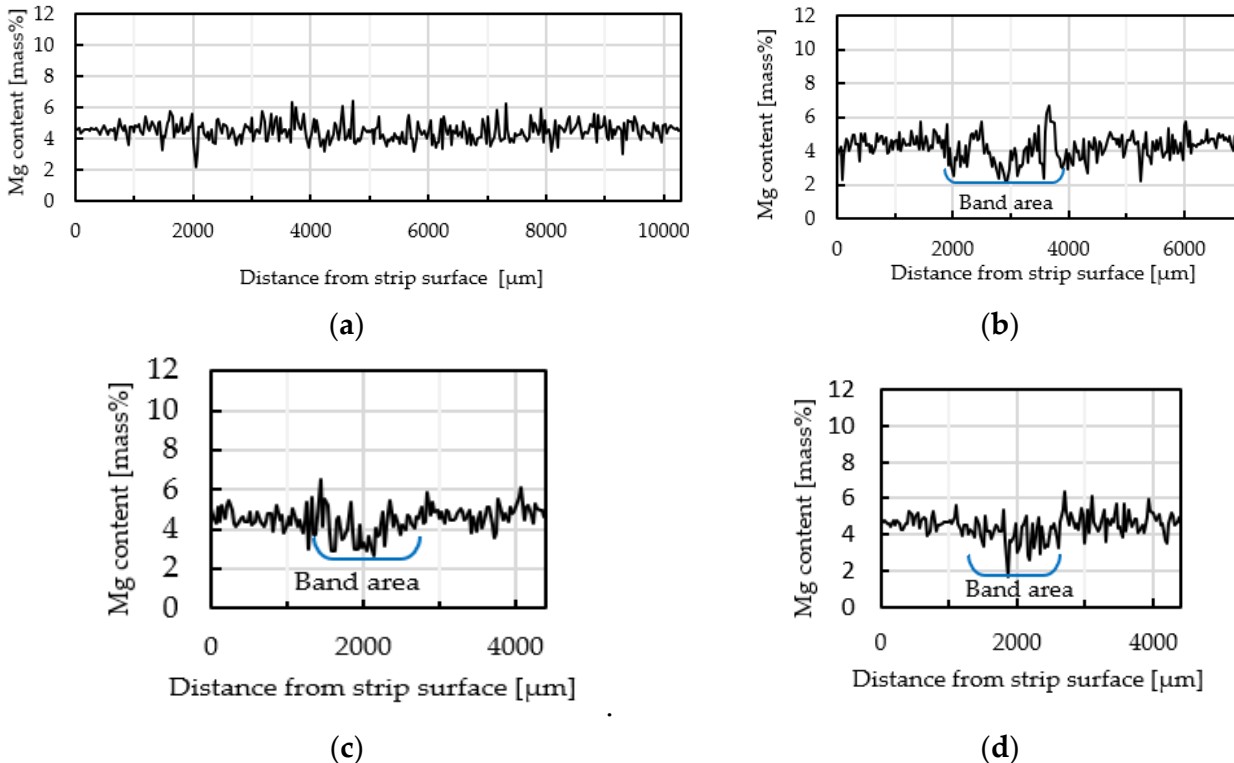

**Figure 10.** Result of EPMA line analysis of Mg. Each section corresponds to a section in Figure 8. (**a**) Section A, (**b**) Section B, (**c**) Section C, and (**d**) Section D.

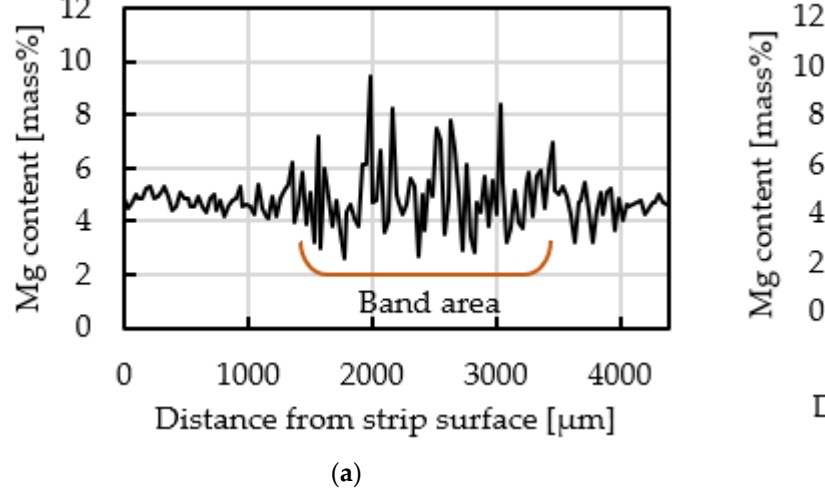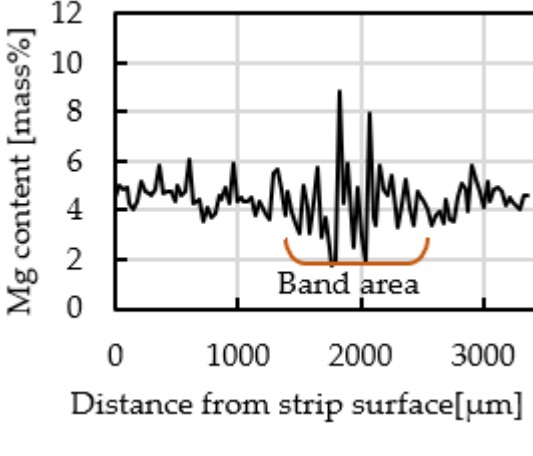

**Figure 11.** *Cont.*

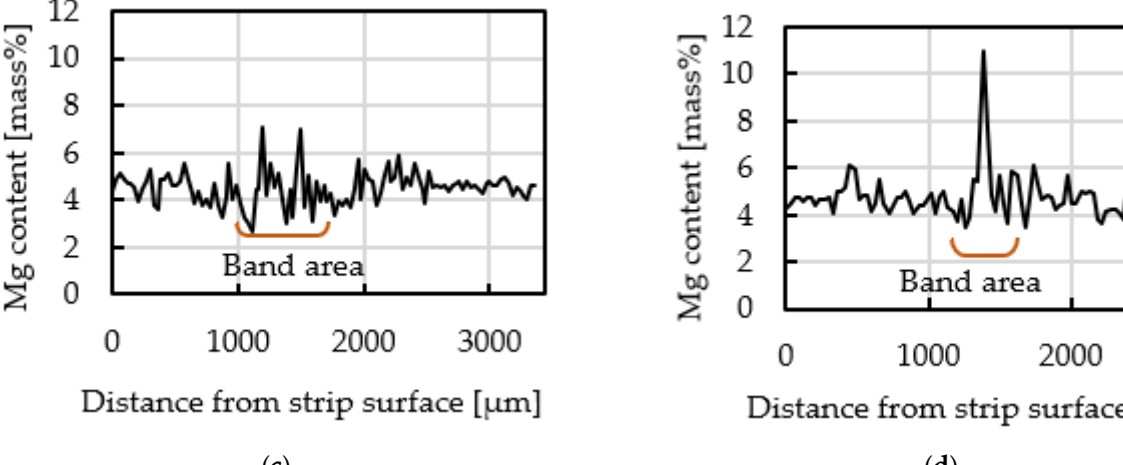

**Figure 11.** Result of EPMA line analysis of Mg. Each section corresponds to a section in Figure 9. (**a**) Section A, (**b**) Section B, (**c**) Section C, and (**d**) Section D.

As-cast strips were cold-rolled to a thickness of 1 mm and annealed. The results of a line analysis of these plates are shown in Figure 12. When the roll load is 2 N/mm, the Mg distribution is not concave; that is, it is lower and almost uniform in the cross section. When the roll load is 2 N/mm, centerline segregation appears to disappear. When the roll load is 88 N/mm, the curve has a convex peak, indicating that centerline segregation remains.

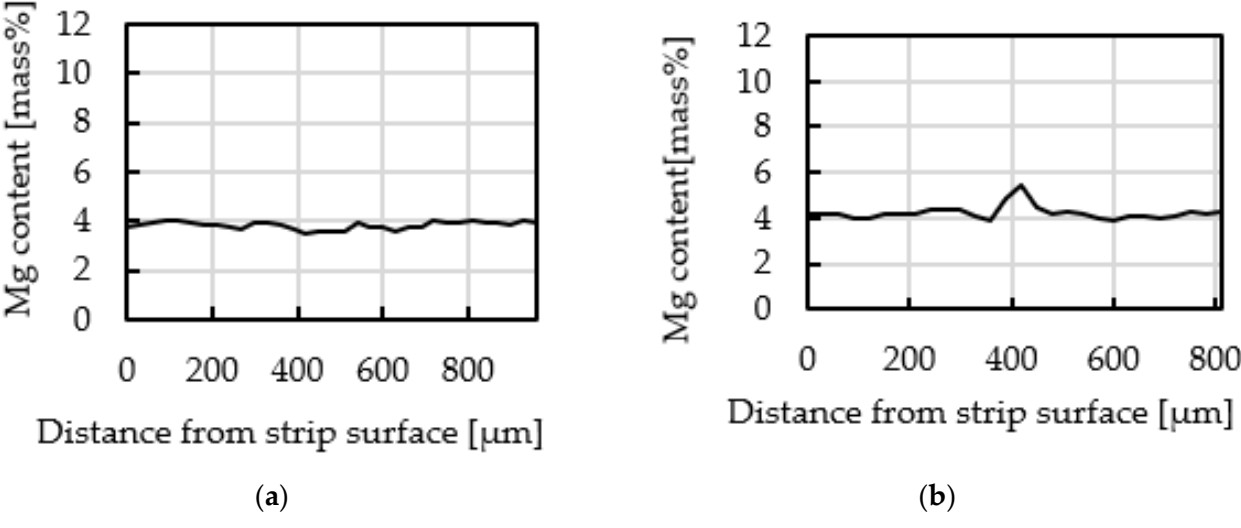

**Figure 12.** Results of EPMA line analysis of strip after cold rolling and annealing. (**a**) Roll load: 2 N/mm. (**b**) Roll load: 88 N/mm.

## 4. Discussion

### 4.1. Tensile Test

When the roll load was 2 N/mm, the tensile strength in the width direction was higher than that in the casting direction, and the elongation in the casting direction was greater than that in the width direction. This reason for these results is not clear at present. When the roll load was 88 N/mm, the tensile strength and elongation in the width direction drastically decreased compared with those in the casting direction, as shown in Figure 5. Cracks could not be detected by penetrant testing after cold rolling, as shown in Figure 4. However, cracks along the casting direction might still be present. Deep drawing tests were conducted to investigate the occurrence of cracks when strain was introduced, and the results are shown in Figure 13. The punch diameter was 32 mm. As-cast strips were

cold-rolled down to a thickness of 1 mm and then annealed at 370 °C for 90 min. When the roll load was 2 N/mm, the surface became rough but did not crack. When the roll load was 88 N/mm, cracks formed in the casting direction but not in the width direction. In the strips cast at a roll load of 88 N/mm, the relationship between cracks in the as-cast strip and cracks in the deep-drawn cup was not clear. There is a strong possibility that cracks in the casting direction, like in the deep-drawn cup, occurred on the tensile test pieces, and as a result, the tensile stress and the elongation in the width direction greatly decreased compared with those in the casting direction. It is thought that cracks might evolve as shown in Figure 14. As shown in Figure 14a, the width of a crack running along the width direction was increased due to cold rolling, and the crack depth decreased, until the crack was finally eliminated. In contrast, as shown in Figure 14b, for cracks running along the casting direction, the crack became longer, and the crack width decreased. At the same time, the crack became oblique and its sidewalls tightly contacted each other, although they did not bond together. The narrowness of the resulting crack means that it cannot be detected by the penetrant test. When the crack is stressed in the direction perpendicular to the casting direction, it opens as shown in Figure 14a. As a result, the tensile strength and elongation in the width direction are lower.

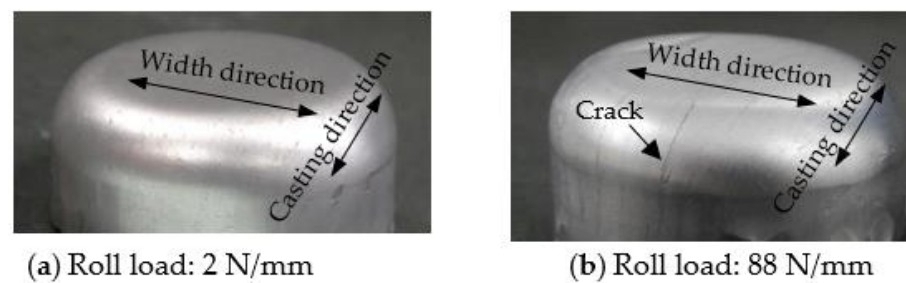

**Figure 13.** Outer surface of cups made by deep drawing. As-cast strip was cold rolled down to 1 mm thick plate and annealed at 370 °C for 90 min.

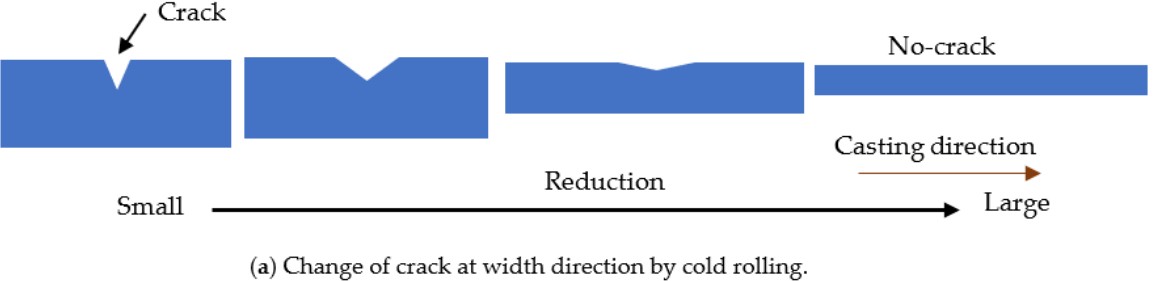

**Figure 14.** Schematic illustration showing transition of a crack's cold rolling.

### 4.2. Dendrite Structure

The roll load affects heat transfer between the roll and the solidified layer, in addition to the amount of squeezed semisolid metal, both of which increase with increasing load. Since metal with a low solid fraction is easily squeezed, the strip thickness decreases as the roll load increases. Therefore, the cooling rate increases with increasing roll load, leading

to a reduction in dendrite size. It can be seen that the dendrites in Figure 9 are finer than those in Figure 8.

A strip cast using a VHSTRC consists of three layers [18,19]. Since the roll load is small, all of the semisolid metal in the center of the strip is not squeezed backward or to the edges, resulting in a band area. As seen in Figure 6, the dendrites became coarse downstream of the roll bite. Similar results are seen in Figure 7. The band area is considered semisolid after the roll bite [21]. The dendrites become coarse due to the latent heat in the band area, because this heat is not drawn by the roll after the roll bite. The solid fraction in the band area decreases as the roll load decreases. Additionally, the latent heat in the band area increases with decreasing roll load. Therefore, the coarsening of the dendrites shown in Figure 8 starts from the roll bite, making them coarser than those in Figure 9.

### 4.3. Mg Distribution

It is clear from Figure 12a that centerline segregation could be almost eliminated by using a small roll load of 2 N/mm. Roll-stop was conducted to investigate the progression of segregation with Weck's reagent. In the roll-stop procedure, the darkest zone exists before the roll bite, where the reduction rate for the thickness of the band area is large, leading to a higher degree of precipitation of Mg from $\alpha$-Al.

As seen in Figures 8b and 9b, especially near section B1, the dendrites become darker outside the band area. This means that the Mg content in the $\alpha$-Al in the band area is lower than in other areas. Section B1 in Figure 9b is darker than section B1 in Figure 8b, indicating a higher Mg content. In Figures 8 and 9, the yellow areas indicated by arrows might be Mg-rich areas. It appears that the segregation of Mg in the band area in Figure 9 is greater than that of Figure 8 because the yellow area is wider in the former case.

Strips cast at roll loads of 2 and 88 N/mm immediately after exiting the roll bite had no ductility, like a strip cooled down to room temperature. Previous research showed that the band area is semisolid at the roll bite [21]. This means that the strips are semisolid immediately after exiting the roll bite. Based on previous research [21,22], the band area is semisolid at a position 30–35 mm before the roll bite. It is thought that the solid fraction in the band area is lower than that at either side of the solidified layer near the roll bite.

The grain boundaries in and around the band area are clear near the roll bite. This means that Mg is precipitated from $\alpha$Al. Clearly, compression due to roll pressure causes Mg precipitation. On either side of the band area, the amount of dissolved Mg in $\alpha$Al decreases as the roll load increases, as determined from Figures 8 and 9, because the grain boundaries in Figure 9 are clearer than those in Figure 8. In the band area, the amount of dissolved Mg in $\alpha$Al decreases as the roll load decreases, as shown in Figures 8 and 9, since the grain boundaries in Figure 8 are clearer than those in Figure 9. After exiting the roll bite, precipitated Mg became dissolved in the $\alpha$Al, as shown by the dark $\alpha$Al changing to bright. The amount of dissolved Mg in the band area in Figure 8 is less than that in Figure 9, and the amount on either side of the band area in Figure 8 is greater than that in Figure 9 after exiting the roll bite. The precipitation of Mg at the roll bite and dissolution of Mg after the roll bite are affected by the roll load, which explains the opposite results shown in Figures 8 and 9. Figure 10b–d show the results of an EPMA line analysis. No segregation of Mg occurs in the band area of the strip cast at a roll load of 2 N/mm. The Mg content at both edges of the band area is higher. The precipitated Mg in the band area does not re-dissolve in the $\alpha$Al but gathers at the band edges, as shown in Figures 8 and 10, when the roll load is 2 N/mm. When the roll load is 88 N/mm, the precipitated Mg from $\alpha$Al on both sides of the band area appears to move into the band area near the roll bite, as indicated by the color on both sides of the band area in Figure 9 and the results of the line analysis in Figure 11. In the band area in Figure 9, Mg-rich areas (yellow areas) are seen after the roll bite. The line analysis shows that the Mg content in the band area is greater than that on either side of the band area. This corresponds to normal segregation of Mg. How the roll load affects this segregation is not clear at this stage. Factors that might be

involved are that the strip was rapidly solidified and became semisolid, and that the band area was super-cooled at the roll bite [21].

Negative segregation in the strip cast at 2 N/mm could be almost eliminated, and normal segregation of the strip cast at 88 N/mm remained after annealing and cold rolling, as shown in Figure 12. These results show that, for high-speed roll casting of an Al-Mg strip, a small roll load of 2 N/mm is useful for eliminating normal segregation of Mg.

## 5. Conclusions

A VHSTRC can cast an Al–5%Mg alloy strip at a speed of 30 m/min. This casting speed is much higher than the casting speed of a CTRCA. The VHSTRC can cast the strip using a much smaller roll load than the CTRCA because the VHSTRC is equipped with rolls made of copper or copper alloy, which has a thermal conductivity much larger than that of the steel roll used for a CTRCA. In a VHSTRC, the roll load is usually larger than 140 N/mm. The effect of roll loads smaller than 140 N/mm on surface cracking and centerline segregation has not been clarified. In this study, the effect of a small roll load on surface cracking and centerline segregation was investigated using an Al–5%Mg alloy because Al–Mg alloy strip castings are prone to these defects.

(1)	When the roll load was 88 N/mm, surface cracking was observed on the as-cast strip, and when the roll load was 2 N/mm, surface cracking was not observed by penetrant testing. The small roll load of 2 N/mm was useful for preventing surface cracking.

(2)	Strips cast at roll loads of 2 and 88 N/mm were cold-rolled down to a thickness of 1 mm. Penetrant testing was conducted, and cracking was not observed on the cold-rolled plates. Tensile testing was conducted on annealed 1 mm thick plates. There was almost no difference in tensile strength and elongation in the casting direction between the plates cast at 2 N/mm and 88 N/mm. However, the tensile strength, and especially elongation, in the width direction for strips cast at 88 N/mm was lower than that of strips cast at 2 N/mm. The cause was determined by deep drawing tests, which revealed cracks in the casting direction that were not detected by penetrant testing.

(3)	In the cast piece obtained by the roll-stop method, the microstructure changed after the strip exited the roll bite. The microstructure in the band area after exiting the roll bite showed that the microstructure became coarser than that in the roll bite. The microstructure on both sides of the band area had an annealed appearance.

(4)	The roll load affected segregation in the band area, judging from the results of a line analysis and etching using Weck's reagent. When the roll load was 2 N/mm, the band area showed no segregation of Mg, and normal segregation of Mg occurred at edges of the band area. When the roll load was 88 N/mm, the band area showed normal Mg segregation. Segregation in the strip cast at 88 N/mm was greater than that at the edges of the band area of the strip cast at 2 N/mm.

(5)	Etching using Weck's reagent showed that the Mg solution state on both sides of the band area of the strip cast at 88 N/mm was different from that cast at 2 N/mm after the roll bite; in particular, the difference was remarkable near the strip surface. The Mg content near the surface of the strip cast at 2 N/mm was greater than that in the strip cast at 88 N/mm.

(6)	When cold rolling and annealing were conducted on as-cast strips, the lack of segregation in the band area for the strip cast at 2 N/mm was eliminated, and normal segregation in the band area of the strip cast at 88 N/mm remained.

(7)	It is concluded that the roll load was dominant for forming surface cracks and centerline segregation, and a small roll load of 2 N/mm was very useful for eliminating surface cracks and centerline segregation.

The roll load of the CTRCA is usually larger than 2 kN/mm, and its roll speed is usually slower than 2 m/min. In this study, the roll load that eliminated surface cracks and centerline segregation was 2 N/mm with a roll speed of 30 m/min. It is clear that the roll load in this study was much smaller than that of a CTRCA. This small roll load was realized using a copper roll with higher thermal conductivity. These results show that using a roll

with higher thermal conductivity and small roll load are the most important factors for realizing high-speed roll casting of strips without surface cracks and centerline segregation.

In this study, Al–5%Mg alloy strips were cast at a roll speed of 30 m/min and roll loads of 2 and 88 N/mm. Roll casting with a small roll load increases the roll life. In addition, the rigidity of the caster frame and rolls can also be decreased, and a lower-power motor can be used. This means that the overall cost of the casting operation can be reduced. Thus, the results of this study demonstrate the practicality of copper and copper alloy rolls in VHSTRCs. However, the upper bound of the roll load and speed that makes defect-free strips is not clear. The effect of alloy elements such as Si, Cu, Fe, and Mn on defect-free roll loads also needs investigation.

**Author Contributions:** Conceptualization, T.H.; methodology, K.Y. and T.H.; validation, K.Y. and T.H.; formal analysis, K.Y. and T.H.; investigation, K.Y. and T.H.; resources, T.H.; data curation, K.Y. and T.H.; writing—original draft preparation, K.Y. and T.H.; writing—review and editing, K.Y. and T.H.; visualization, K.Y. and T.H.; supervision, T.H.; project administration, T.H.; funding acquisition, T.H. All authors have read and agreed to the published version of the manuscript.

**Funding:** SUZUKI FOUNDATION, https://www.suzukifound.jp.

**Institutional Review Board Statement:** Not applicable.

**Informed Consent Statement:** Not applicable.

**Conflicts of Interest:** The authors declare no conflict of interest.

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
