# Peer review of "High-Speed and Low-Load Twin-Roll Casting of Al–5%Mg Strip"

_metals, doi:10.3390/met13010072_

Round 1

Reviewer 1 Report

The present manuscript explores the effect of a small roll load on surface cracking and centerline segregation of Al-5%Mg alloy. It is recognized that the roll load plays a dominant role on the formation surface cracks and centerline segregation, and a small roll load of 2 N/mm is very useful for eliminating surface cracks and centerline segregation. This is a well-organized manuscript which is worth for publication.

Author Response

Thank you for your helpful comments.

Reviewer 2 Report

1)Sufficient literature is available in this area of work, and the introduction section needs to be expanded with more citations.

2) Please specify the test standards adopted for the tensile test.

3) Correct the spellings in "Figure 4. Surafeces of stripd "

4) Section 3.2: In figure 5, casting direction is spelt as cating direction. Make corrections.

5) Section 3.2: Elaborate discussion is required for the increase in tensile strength with suitable citations

6) In Figure 5, annotation is spelled as nand.

7) The reason for the variation in dendritic structure is not discussed. 

8) Kindly annotate 6b and 7b mentioning the nature of the microstructure obtained.

9) The presence of intermetallics,  α Mg / α+γ eutectic, is not discussed. 

10) The mode of crack propagation is to be discussed.

Author Response

Thank you for your helpful comments and suggestions.

I have revised the manuscript in accordance with your recommendations.

Please review the revised manuscript.

Reviewer 2

Comments and Suggestions for Authors

  • Sufficient literature is available in this area of work, and the introduction section needs to be expanded with more citations.

Answer: Thank you for suggestion. Additional references were added to the Introduction. Revised sentences are shown in green. 

  • Please specify the test standards adopted for the tensile test.

Answer: The test piece was small, and the tensile tests did not conform to standards such as ASTM or JIS.

  • Correct the spellings in "Figure 4. Surafeces of stripd "

Answer: Thank you. I changed "Figure 4. Surfaeces of stripd " to "Figure 4. Surfaces of strips ".

  • Section 3.2: In figure 5, casting direction is spelt as cating direction. Make corrections.

Answer: Thank you. I corrected “cating” to “casting”.

  • Section 3.2: Elaborate discussion is required for the increase in tensile strength with suitable citations

Answer: It is not clear that the tensile strength in the width direction was greater than that in the casting direction. We added this in the Discussion. (All changes shown in green characters)

6) In Figure 5, annotation is spelled as nand.

Answer: Thank you. I corrected “nand” to “and”.

  • The reason for the variation in dendritic structure is not discussed.

Answer: We added a discussion of this (green characters).

  • Kindly annotate 6b and 7b mentioning the nature of the microstructure obtained.

Answer: We added “A1-D3 in (b) correspond to A1-D3 in (a) “to Figures 6 and 7.

The presence of intermetallics, α Mg / α+γ eutectic, is not discussed.

Answer:  Thank you for this valuable suggestion. The presence of intermetallics, such as α Mg / α+γ eutectic, was not investigated. This is a topic for future research.

  • The mode of crack propagation is to be discussed.

Answer: Fracture is outside the scope of this study, so we do not discuss the crack propagation mode.

Reviewer 3 Report

The manuscript deals with defects due to cracking and the quality of the surface in twin roll casting with increased casting speed. The results are in principle of practical and scientific interest. However, it is considered necessary to fundamentally revise the manuscript:

(a) The authors speak at the beginning about investigating the influence of the rolling force on the above appearances. However, this force is ultimately a resulting variable from the setting of the rolling gap and the sheet thickness resulting from it, which is mentioned by the authors in the course of the manuscript. This means that the set rolling gap or the sheet thickness produced is actually the varying variable and the rolling force is a resulting variable. Furthermore, the thicknesses investigated differ by almost 20 %, which in turn is likely to have an effect on thermally induced microstructural changes and must be considered. It therefore seems more reasonable to speak throughout the manuscript of the influence of the sheet thickness produced rather than the influence of the force and to treat the latter as a resulting variable.

(b) The state of the literature needs to be addressed in more depth. In particular, the consideration of current sources that deal intensively with crack formation in twin roll casting is missing, e.g. 10.1007/s11665-021-05530-9. Furthermore, of the 17 references listed, 8 are publications by the authors, which should be reduced.

(c) Chapter 4.1 should be removed. The assumptions presented on the formation of cracks are not supported by measurements or model-based calculations and only represent a hypothesis. This hypothesis can be reduced to a few sentences and formulated as an outlook for further investigations. The authors should instead focus on the useful findings they have obtained from microstructural analyses and mechanical material tests.

(d) Further points:

Lines 37 ff: To which examinations does "The rolls were ..." refer?

Lines 46 ff.: If the temperature of the copper rolls is lower, then the temperature difference is also lower and thus the amount of heat transferred. The sentence should be revised.

Lines 53 ff: Why are the deformation load and roll load lower with copper rolls than with steel rolls?

Lines 58 ff: The statement that higher speeds are possible with copper rollers must be better substantiated.

Line 89: For which tests were the two roll diameters 300 and 50 mm used? This is not discussed in the further descriptions.

Aluminium must be listed in Table 1.

Line 122: Why was annealing used?

Line 152: It is surprising that the cracks are eliminated by a cold rolling process. Was the depth of the cracks investigated?

Line 297: A different terminology should be chosen here. The "thicker" grain boundaries are a higher proportion of precipitations presumably.

Figure 11: "Section" instead of "Sectiom" and "scetcons".

Lines 567 ff: Strains occur both in the tensile tests described above and in deep drawing. The sentence should be revised.

"Weck's reagent" is mentioned more than ten times in the text. To mention it one times is sufficient.

Author Response

Thank you for your helpful comments and suggestions.

I have revised the manuscript in accordance with your recommendations.

Please review the revised manuscript.

Reviewer 3

(a) The authors speak at the beginning about investigating the influence of the rolling force on the above appearances. However, this force is ultimately a resulting variable from the setting of the rolling gap and the sheet thickness resulting from it, which is mentioned by the authors in the course of the manuscript. This means that the set rolling gap or the sheet thickness produced is actually the varying variable and the rolling force is a resulting variable. Furthermore, the thicknesses investigated differ by almost 20 %, which in turn is likely to have an effect on thermally induced microstructural changes and must be considered. It therefore seems more reasonable to speak throughout the manuscript of the influence of the sheet thickness produced rather than the influence of the force and to treat the latter as a resulting variable.

Answer: Thank you for your important suggestion.

The strip thickness affects the cooling rate and microstructure. When the effect of only the thickness is investigated, the thickness is controlled by the solidification time, which is determined by the roll speed or solidification length. The use of the solidification length is appropriate as the heat transfer between the roll and molten metal is almost constant. When the roll speed is used, the heat transfer changes with roll speed. The roll load is kept constant when the effect of the strip thickness on the microstructure is investigated because the roll load affects the heat transfer and the amount of squeezing of the semisolid metal. In this study, the roll speed and solidification length were constant and the roll load was changed. The heat transfer between the roll and solidified layer is affected by the roll load. The amount of squeezed semisolid metal is affected by the roll load. These cannot be discussed in terms of the effect of the strip thickness. The strip thickness decreases as the roll load increases, as you suggested. Taking everything into consideration, we used the roll load.   

(b) The state of the literature needs to be addressed in more depth. In particular, the consideration of current sources that deal intensively with crack formation in twin roll casting is missing, e.g. 10.1007/s11665-021-05530-9. Furthermore, of the 17 references listed, 8 are publications by the authors, which should be reduced.

Answer: Thank you for kind suggestion. I added references and improved the text. Revised sentences are shown in green. Some references to our papers were removed.

 (c) Chapter 4.1 should be removed. The assumptions presented on the formation of cracks are not supported by measurements or model-based calculations and only represent a hypothesis. This hypothesis can be reduced to a few sentences and formulated as an outlook for further investigations. The authors should instead focus on the useful findings they have obtained from microstructural analyses and mechanical material tests.

Answer: In accordance with your recommendation, Chapter 4.1 has removed.

(d) Further points:

Lines 37 ff: To which examinations does "The rolls were ..." refer?

Answer: This is commonly said. There is not references. “The rolls were made from tool steel for hot working to conduct the hot rolling as the de-formation stress was large at high temperature.” is changed to “CTRCA rolls are made from steel.”

Lines 46 ff.: If the temperature of the copper rolls is lower, then the temperature difference is also lower and thus the amount of heat transferred. The sentence should be revised.

Answer: The sentence was revised as follows, and appears in blue in the revised paper

The thermal conductivity of the copper roll is about eight times larger than that of steel, and increasing temperature of the copper roll is lower than that of the steel roll during roll-solidification layer (including a strip) contacting. The heat transfer between the copper roll and the solidification layer becomes larger than that for the steel roll as the temperature of the copper roll surface is lower than that of the steel roll.

Lines 53 ff: Why are the deformation load and roll load lower with copper rolls than with steel rolls?

Answer: Thank you for suggestion. This sentence is unnecessary and inappropriate, and was therefore removed.

Lines 58 ff: The statement that higher speeds are possible with copper rollers must be better substantiated.

Answer: A reference has added with a reference number of “9”.

Line 89: For which tests were the two roll diameters 300 and 50 mm used? This is not discussed in the further descriptions.

Answer: Casting in this study was conducted using a roll with a 300 mm diameter and a 50 mm width.

Aluminium must be listed in Table 1.                           

Answer: “Al” has been added to Table 1.

Line 122: Why was annealing used?

Answer: Annealed Al-Mg plate is usually used for sheet forming. Segregation is somewhat reduced (improved) by annealing, so it was thought that the tensile testing should be investigated.   

Line 152: It is surprising that the cracks are eliminated by a cold rolling process. Was the depth of the cracks investigated?

Answer: Thank you. This expression is not appropriate and it was changed as below. “Cracks on the plate surfaces after cold rolling could not be detected by penetrant testing.” Also, further discussion has been added.

 The depths of all cracks could not be measured because of the large number present. We used penetrant testing to check the existence of cracks on the strip surface. The crack depth was up to 1 mm. When cracks were detected, the results of tensile testing and deep drawing were worse.

Line 297: A different terminology should be chosen here. The "thicker" grain boundaries are a higher proportion of precipitations presumably.

Answer: “the grain boundaries are thicker” is revised to “there is a higher density of precipitates at grain boundaries”.

Figure 11: "Section" instead of "Sectiom" and "scetcons".

Answer: “section” and “sections” are changed to “section”.

Lines 567 ff: Strains occur both in the tensile tests described above and in deep drawing. The sentence should be revised.

Answer: This sentence was revised below. “Cracks could not be detected by penetrant testing after cold rolling, as shown in Figure 4. However, cracks along the casting direction might still be present”.

"Weck's reagent" is mentioned more than ten times in the text. To mention it one times is sufficient.

Answer: Some instances of "Weck's reagent" were removed.

Thank you for your helpful comments and suggestions.

I have revised the manuscript in accordance with your recommendations.

Please review the revised manuscript.

Reviewer 3

(a) The authors speak at the beginning about investigating the influence of the rolling force on the above appearances. However, this force is ultimately a resulting variable from the setting of the rolling gap and the sheet thickness resulting from it, which is mentioned by the authors in the course of the manuscript. This means that the set rolling gap or the sheet thickness produced is actually the varying variable and the rolling force is a resulting variable. Furthermore, the thicknesses investigated differ by almost 20 %, which in turn is likely to have an effect on thermally induced microstructural changes and must be considered. It therefore seems more reasonable to speak throughout the manuscript of the influence of the sheet thickness produced rather than the influence of the force and to treat the latter as a resulting variable.

Answer: Thank you for your important suggestion.

The strip thickness affects the cooling rate and microstructure. When the effect of only the thickness is investigated, the thickness is controlled by the solidification time, which is determined by the roll speed or solidification length. The use of the solidification length is appropriate as the heat transfer between the roll and molten metal is almost constant. When the roll speed is used, the heat transfer changes with roll speed. The roll load is kept constant when the effect of the strip thickness on the microstructure is investigated because the roll load affects the heat transfer and the amount of squeezing of the semisolid metal. In this study, the roll speed and solidification length were constant and the roll load was changed. The heat transfer between the roll and solidified layer is affected by the roll load. The amount of squeezed semisolid metal is affected by the roll load. These cannot be discussed in terms of the effect of the strip thickness. The strip thickness decreases as the roll load increases, as you suggested. Taking everything into consideration, we used the roll load.   

(b) The state of the literature needs to be addressed in more depth. In particular, the consideration of current sources that deal intensively with crack formation in twin roll casting is missing, e.g. 10.1007/s11665-021-05530-9. Furthermore, of the 17 references listed, 8 are publications by the authors, which should be reduced.

Answer: Thank you for kind suggestion. I added references and improved the text. Revised sentences are shown in green. Some references to our papers were removed.

 (c) Chapter 4.1 should be removed. The assumptions presented on the formation of cracks are not supported by measurements or model-based calculations and only represent a hypothesis. This hypothesis can be reduced to a few sentences and formulated as an outlook for further investigations. The authors should instead focus on the useful findings they have obtained from microstructural analyses and mechanical material tests.

Answer: In accordance with your recommendation, Chapter 4.1 has removed.

(d) Further points:

Lines 37 ff: To which examinations does "The rolls were ..." refer?

Answer: This is commonly said. There is not references. “The rolls were made from tool steel for hot working to conduct the hot rolling as the de-formation stress was large at high temperature.” is changed to “CTRCA rolls are made from steel.”

Lines 46 ff.: If the temperature of the copper rolls is lower, then the temperature difference is also lower and thus the amount of heat transferred. The sentence should be revised.

Answer: The sentence was revised as follows, and appears in blue in the revised paper

The thermal conductivity of the copper roll is about eight times larger than that of steel, and increasing temperature of the copper roll is lower than that of the steel roll during roll-solidification layer (including a strip) contacting. The heat transfer between the copper roll and the solidification layer becomes larger than that for the steel roll as the temperature of the copper roll surface is lower than that of the steel roll.

Lines 53 ff: Why are the deformation load and roll load lower with copper rolls than with steel rolls?

Answer: Thank you for suggestion. This sentence is unnecessary and inappropriate, and was therefore removed.

Lines 58 ff: The statement that higher speeds are possible with copper rollers must be better substantiated.

Answer: A reference has added with a reference number of “9”.

Line 89: For which tests were the two roll diameters 300 and 50 mm used? This is not discussed in the further descriptions.

Answer: Casting in this study was conducted using a roll with a 300 mm diameter and a 50 mm width.

Aluminium must be listed in Table 1.                           

Answer: “Al” has been added to Table 1.

Line 122: Why was annealing used?

Answer: Annealed Al-Mg plate is usually used for sheet forming. Segregation is somewhat reduced (improved) by annealing, so it was thought that the tensile testing should be investigated.   

Line 152: It is surprising that the cracks are eliminated by a cold rolling process. Was the depth of the cracks investigated?

Answer: Thank you. This expression is not appropriate and it was changed as below. “Cracks on the plate surfaces after cold rolling could not be detected by penetrant testing.” Also, further discussion has been added.

 The depths of all cracks could not be measured because of the large number present. We used penetrant testing to check the existence of cracks on the strip surface. The crack depth was up to 1 mm. When cracks were detected, the results of tensile testing and deep drawing were worse.

Line 297: A different terminology should be chosen here. The "thicker" grain boundaries are a higher proportion of precipitations presumably.

Answer: “the grain boundaries are thicker” is revised to “there is a higher density of precipitates at grain boundaries”.

Figure 11: "Section" instead of "Sectiom" and "scetcons".

Answer: “section” and “sections” are changed to “section”.

Lines 567 ff: Strains occur both in the tensile tests described above and in deep drawing. The sentence should be revised.

Answer: This sentence was revised below. “Cracks could not be detected by penetrant testing after cold rolling, as shown in Figure 4. However, cracks along the casting direction might still be present”.

"Weck's reagent" is mentioned more than ten times in the text. To mention it one times is sufficient.

Answer: Some instances of "Weck's reagent" were removed.

Round 2

Reviewer 3 Report

The reviewer thanks the authors for the consideration of the recommendation of the first review. The manuscript is suggested for publication.